# Heritability and Genetic Advance Estimates of Key Shea Fruit Traits

Wisdom Edem Anyomi [1,*], Michael Teye Barnor [1], Agyemang Danquah [2], Kwadwo Ofori [2], Francis Kwame Padi [3], Silas Wintuma Avicor [3], Iago Hale [4] and Eric Yirenkyi Danquah [2]

1. Cocoa Research Institute of Ghana, Bole Sub-Station, Bole P.O. Box BL 41, Ghana
2. West Africa Center for Crop Improvement, School of Agriculture, College of Basic and Applied Sciences, University of Ghana, Accra PMB 30, Ghana
3. Cocoa Research Institute of Ghana, New Tafo-Akim P.O. Box 8, Ghana
4. Department of Agriculture, Nutrition, and Food Systems, University of New Hampshire, Durham, NH 03824, USA
* Correspondence: lordedems@gmail.com

**Abstract:** Genetic erosion of shea trees, which has been on-going at an alarming rate, has necessitated urgent conservation attentions. Owing to the vast geographical distribution of the species across Ghana, in situ germplasms conservation was established by tagging and monitoring selected trees annually. Technologies have been developed that enable shea germplasms to be grafted, allowing for the development of germplasm banks at the research station of the Cocoa Research Institute of Ghana, Bole. However, before these materials could be used in crop improvement programs, there is a need to evaluate them for key fruit traits relevant to the global shea business. This experiment was carried out to evaluate the tagged in situ shea trees for fruit and nut traits. Freshly harvested shea fruits were evaluated for their brix, pulp yield and kernel size properties to see if there was the needed diversity for crop improvement gains. Eight key traits were studied, with all showing significant difference, with high broad sense heritability and genetic advance for all the traits, indicating the potential for genetic gains in breeding programs. Qualitative analysis classified the fruits into five shapes, ellipsoid fruit shape was the most frequent observation (69.5%), while oblong was the least represented (1%). Fruit surface pubescence indicated that the surfaces without hairs (smooth) were slightly higher in number (52.6%) than the surface with hairs (rough), which were 47.4%. Pearson correlation studies showed a positive significant relationship between kernel weight and fruit weight (0.68), fruit length (0.48), fruit width (0.51), pulp weight (0.5) and shell weight (0.77). Key components responsible for total variations observed were decomposed from the first two principal components (PC), which cumulatively explained 78.4% of the total observed variation within the materials. PC1 alone contributed 46.4%, while PC2 contributed 32%. Fruit weight, fruit length, fruit width, pulp weight, nut weight, shell weight and kernel weight were contributing traits to variations observed in PC1, while brix and percent pulp contributed to the variations observed in PC2. Percent kernel to nut ratio contributed to the variations observed in PC3. Clustering of the germplasms showed no regular pattern based on location or any particular trait, indicating a high level of diversity at 58% of the Pearson dissimilarity index.

**Keywords:** heritability; genetic advance; shea; multivariate; phenotype; traits

## 1. Introduction

The shea butter tree, known scientifically as *Vitellaria paradoxa* C.F. Gaertner, is a deciduous tree indigenous mainly to West Africa and parts of Eastern Africa, especially Uganda. It is an underdeveloped oilseed species, only second to oil palm in importance in Sub-Sahara Africa [1]. The global shea butter market is projected to hit $4 billion by 2032 [2]. The tree produces fruits once a year and these fruits develop from its insect-pollinated hermaphrodite inflorescence, with bees cited as the main pollinator [3]. Shea

fruit development takes between four and six months [4] and coincidentally ripens in the rainy season, a situation that normally negatively affects kernel quality due to inadequate sunshine for drying. When fully mature, a tree can produce between 15–20 kg of fruits, which will lead to about 5 kg of butter on processing [5]. The fruits produced by shea trees vary in size, shape, appearance and taste [4,6,7]; while some fruits appear smooth, others appear with pubescence on their surface. The ripened fruit color also varies from a shade of green to yellow [7]. Surrounding the fruit is normally a thick mesocarp, the pulp of which is sweet and nutritious, providing energy when consumed [8]. These ripened fruits serve some important socioeconomic purposes, such as hunger alleviation and contribution to household incomes during the gathering season [8,9]. Indeed, the hunger alleviation potential of shea pulp has drawn some international recognition as a food security crop [10]. Reported to contain up to 34.6% of the total dry matter, 21–25% being total free sugars and 14.5% reducing sugars [11], shea pulp can be used in commercial production of wines, jams, ethanol and vinegar [12]. Shea's importance is furthermore highlighted through foreign trade as a result of its unique fatty acid content and tocopherol range, which makes it particularly suitable in bakery, pharmaceutical and cosmetic industries globally [13]. Variations have been reported on fruit size, pilosity, sweetness and color, among trees within some communities [14] and these variations have also been found within different agro-ecologies and populations [14]. However, the tree remains undomesticated and there have not been any suggestions on how these traits are inherited and can be improved in breeding programs.

In order to domesticate shea trees, several propagation techniques have been developed [15–18] and the most efficient is grafting [17,18]. This provides the opportunity to conserve trees of interest when they are identified. It has enabled the development of shea parklands for interested communities and a germplasm bank for shea trees. Germplasm is needed as a repository to provide genetic materials, traits and diversity needed for shea tree varietal development. To prevent duplication and the conservation of unwanted resources, there is a need to evaluate their diversity and measure their genetic parameters, characterizing them by traits. The purpose of this study was to evaluate the fruit and kernel diversity of the tagged materials, identify the key components contributing to their variation and to estimate the heritability and genetic advancement of the assembled germplasm to ascertain if gains will be made when they are used in breeding programs.

## 2. Materials and Methods

### 2.1. Plant Material and Study Area

Trees were selected by first identifying communities with shea populations. Trees were randomly selected from these populations, multiple trees selected within the same population were 50 m apart. A total of 95 trees were sampled for this study. Fifteen ripened shea fruits were randomly collected per tree between May and June when trees were fruiting. These were sent to the laboratory of the Cocoa Research Institute of Ghana, Bole substation for fruit measurements. Fruits were collected in five regions in Ghana (Table 1), namely Upper East, the part that is near the Sudan savannah agroecology, Northern, Upper West and Savannah regions, the Guinea savannah agroecology where shea is mostly found and Bono regions, which is in the forest transitional agroecology.

**Table 1.** List of germplasms collected and their respective regions of collection.

| ID | Location | ID | Location | ID | Location |
|----|----------|----|----------|----|----------|
| CRIG 189 | Bawku | CRIG 104 | Bole | CRIG KA 87 | Bole |
| G6 | Bawku | CRIG 105 | Bole | CRIG KA 24 | Bole |
| G2 | Bawku | CRIG R1NBT3 | Bole | CRIG KA 21 | Bole |
| CRIG PHBA 43 | Bole | CRIG R2CT1 | Bole | CRIG 91 | Damongo |
| CRIG PHBA 54 | Bole | CRIG R2CT2 | Bole | CRIG 90 | Damongo |
| CRIG PHBA 28 | Bole | CRIG R1ENT4 | Bole | CRIG 293 | Damongo |
| CRIG PHBA 50 | Bole | CRIG R2EBT1 | Bole | CRIG 84 | Damongo |

**Table 1.** *Cont.*

| ID | Location | ID | Location | ID | Location |
|---|---|---|---|---|---|
| CRIG PHBA 37 | Bole | CRIG R2NBT1 | Bole | CRIG 86 | Damongo |
| CRIG PHBA 51 | Bole | CRIG KRAAL | Bole | CRIG 85 | Damongo |
| CRIG PHBA 25 | Bole | CRIG J48 | Bole | CRIG 95 | Kintampo |
| CRIG P1R1T5 | Bole | CRIG J47 | Bole | CRIG 94 | Kintampo |
| CRIG EA 1 | Bole | CRIG GMSA | Bole | CRIG 93 | Kintampo |
| CRIG EA 4 | Bole | CRIG MB 13 | Bole | CRIG 15 | Kintampo |
| CRIG RH 1 | Bole | CRIG MB 16 | Bole | CRIG 17 | Kintampo |
| CRIG LAB C | Bole | CRIG MB 5 | Bole | CRIG 64 | Kintampo |
| CRIG SG130 | Bole | CRIG MB 14 | Bole | CRIG 169 | Navrongo |
| CRIG SG128 | Bole | CRIG KA 30 | Bole | CRIG 172 | Navrongo |
| CRIG SG170 | Bole | CRIG KA 11 | Bole | CRIG 39 | Tamale |
| CRIG SG129 | Bole | CRIG KA 105 | Bole | CRIG 510 | Wa |
| CRIG SG171 | Bole | CRIG KA 16 | Bole | CRIG 130 | Wa |
| CRIG SG302 | Bole | CRIG KA 09 | Bole | CRIG 125 | Wa |
| CRIG SG118 | Bole | CRIG KA 93 | Bole | CRIG 123 | Wa |
| CRIG SG254 | Bole | CRIG KA 1 | Bole | CRIG 136 | Wa |
| CRIG SG97 | Bole | CRIG KA 27 | Bole | CRIG 138 | Wa |
| CRIG SG100 | Bole | CRIG KA 110 | Bole | CRIG 126 | Wa |
| CRIG SG116 | Bole | CRIG KA 29 | Bole | CRIG 141 | Wa |
| CRIG SG282 | Bole | CRIG KA 3 | Bole | CRIG 137 | Wa |
| CRIG SG114 | Bole | CRIG KA 10 | Bole | CRIG 59 | Walewale |
| CRIG SG142 | Bole | CRIG KA 2 | Bole | CRIG 330 | Walewale |
| CRIG SG284 | Bole | CRIG KA 5 | Bole | CRIG 4 | Yendi |
| CRIG SG113 | Bole | CRIG KA 33 | Bole | CRIG 6 | Yendi |
| CRIG 107 | Bole | CRIG KA 40 | Bole | | |

*2.2. Data Collection*

Figure 1 shows the various shapes of shea fruits and dry kernels according to the shea descriptor [19]. Figure 2 shows a cross section of shea fruits with intact nuts and yellow fleshy pulp.

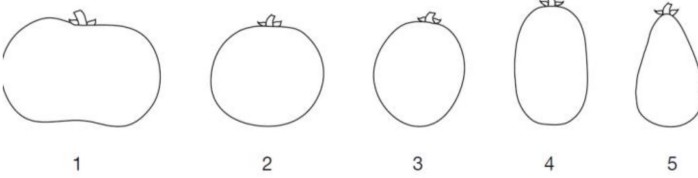

```
1   Oblate
2   Spheroid
3   Ellipsoid
4   Oblong
5   Ovoid
99  Other (i.e. 'irregular' specify in descriptor 7.6 Notes)
```

**Figure 1.** Various shapes of shea kernel (INIA, 2006).

The following parameters were recorded:

Fresh weight was measured as the weight of intact fruit as collected on ripening with an electronic scale. Length of fruit was recorded as the length of the longest section of the fruit, from the point of attachment to the pedicel to the apex, while width of fruit was measured as the widest girth of the middle section of the fruit as, shown in Figure 3, and was performed with the aid of calipers. The picture also depicts how a fully ripened shea fruit looks when it is cut open, with the nut surrounded by a yellow sweet pulp. On the far right is a kernel after it has been dried.

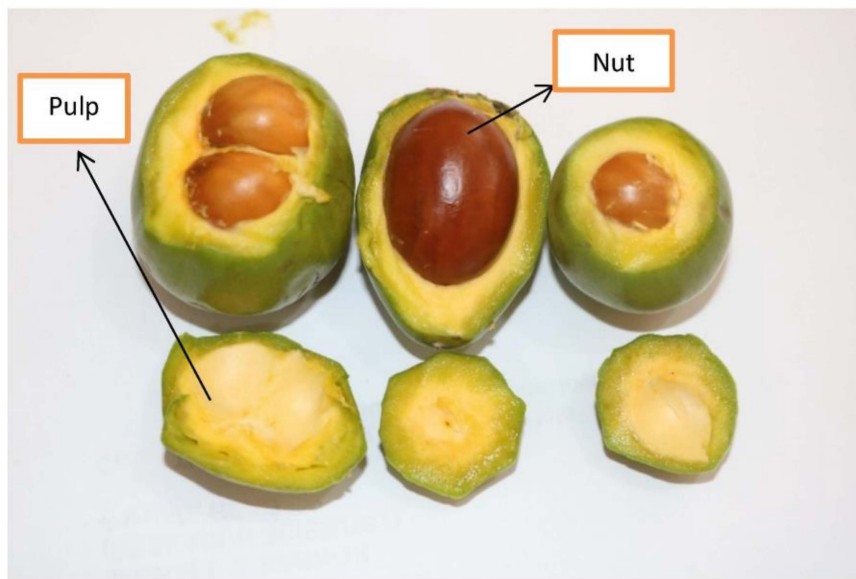

**Figure 2.** Cross section of shea fruits.

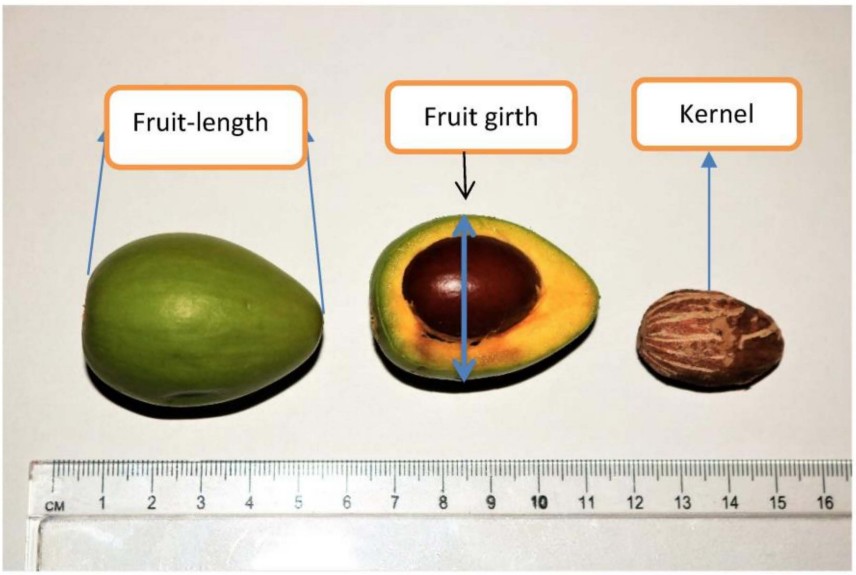

**Figure 3.** Fresh shea fruit and kernel dimension.

Shape 5 in Figure 1 is ovoid with the pedicel attachment point at the narrow end of the fruit. In our observation, the shape in Figure 4, with the pedicel at the broader end, was largely what we found in our population. Pubescence on shea appear as brown dense hair, which is easily removed by swiping the fingers over the surface.

Brix is a measure of the total sugars contained in the pulp covering the ripe fruit; it was measured with a digital refractometer. Pulp weight is the weight of the total flesh covering the fruit upon ripening, measured with a scale. Dry weight is the weight of the dried nut and shell weight is the weight of the dried shell covering the kernel.

Kernel weight is regarded as the most important component, which contains the butter, and was measured with a digital scale after being oven dried at 45 °C for 3 days.

Pulp to fruit weight was measured as the pulp weight divided by the fresh intact fruit weight. Kernel to fresh weight ratio is a measure of the dried kernel as a fraction of the total fresh fruit weight. It indicates the proportion of total fruit weight that is due to the main kernel.

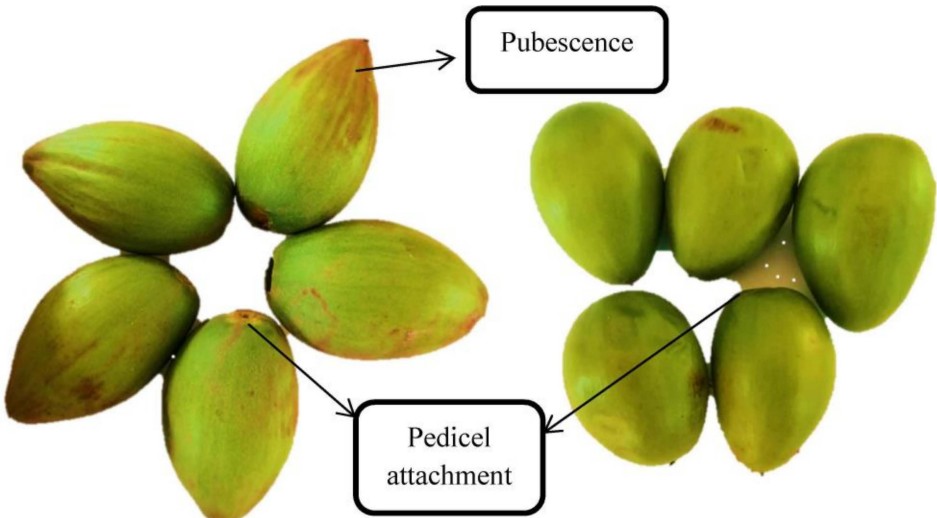

**Figure 4.** Inverted ovoid shea fruits showing pedicel attachment point and pubescence.

Nut to fruit weight ratio describes the dry nut weight as a factor of fresh fruit weight and it indicates the extra weight carried by pickers because of fresh fruits; this is normally discarded after depulping and can be used to measure effective work done. Kernel to nut ratio, also referred to as shelling percentage, describes the weight of dried nut that is due to the kernel itself.

### 2.3. Data Analysis

Statistical analysis of variance was carried out with Genstat version 12 and variance components were extracted for heritability estimates using the "variability" package in R statistical software [20]. Pearson correlation was used to determine the associations among the traits measured. Principal component analysis (PCA) was used to determine the Percent contribution of the measured traits to the total observed variation in the study. Correlations and clustering of genotypes were based on UPGMA using XLSTAT software.

Analysis of variance followed the following terms:

| Source | D.F | Mean Square | Expected Mean Square |
|---|---|---|---|
| Replications | $(r-1)$ | $M_1$ | $\sigma_e^2 + g\,\sigma_r^2$ |
| Genotype | $(g-1)$ | $M_2$ | $\sigma_e^2 + r\,\sigma_g^2$ |
| Error | $(r-1)(g-1)$ | $M_3$ | $\sigma_e^2$ |

where,

$r$ = Number of replications;

$g$ = Number of genotypes;

$\sigma_e^2,\ \sigma_r^2,\ \sigma_g^2$ = Variances due to error, replications and genotypes, respectively.

$M_1,\ M_2,\ M_3$ = Mean squares for replication, genotypes and error, respectively

The standard error for differences between treatment means was calculated from ANOVA table:

$$\text{S.Em} = \sqrt{\frac{M_3}{r}}$$

where,

S.Em = Standard error of mean;

$M_3$ = Error mean square;

$r$ = Number of replications.

The coefficient of variation (CV) was calculated by using the following formula:

Variability parameters

$$CV = \frac{\sqrt{M_3}}{\overline{\overline{X}}} \times 100$$

where,

CV = Coefficient of variation;

$M_3$ = Error mean square;

$\overline{X}$ = General mean.

Mean was computed by dividing the sum of all observations in a sample by their total number. Thus,

$$\overline{X} = \frac{\sum X_{ij}}{n}$$

where:

$\overline{X}$ = Population mean;

$X_{ij}$ = Any observation in *j*th genotype in *i*th replication;

*n* = Number of observations;

Range = The difference between the lowest and the highest value for each character.

The genotypic, phenotypic and environmental components were estimated, as explained by [21].

Genotypic variance ($\sigma_g^2$). The variance contributed by the genetic causes:

$$\sigma_g^2 = \frac{M_2 - M_3}{r}$$

where:

$\sigma_g^2$ = Genotypic variances;

$M_3$, $M_2$ = Mean squares for error and genotypes, respectively *r* = Number of replication.

Environmental variance

Defined as error mean square due to environmental variances

$$\sigma_e^2 = M_3$$

where,

$\sigma_g^2$ = Environmental variances

$M_3$ = Error mean square

Phenotypic variance

The sum of the variances contributed by genetic causes and environmental factors, and was calculated as follows,

$$\sigma_p^2 = \sigma_e^2 + \sigma_g^2$$

where,

$\sigma_p^2, \sigma_e^2, \sigma_g^2$ = Variances due to phenotype, error and genotype, respectively.

Coefficient of variation

The coefficients of phenotypic and genotypic variations were calculated by the formula suggested by [22].

Phenotypic coefficient of variation (PCV)

$$PCV(\%) = \frac{\sqrt{\sigma_p^2}}{\overline{X}} \times 100$$

Genotypic coefficient of variation (GCV)

$$GCV(\%) = \frac{\sqrt{\sigma_g^2}}{\overline{X}} \times 100$$

GCV and PCV were categorized as low, moderate and high by following [23]:

0–10%: Low

10–20%: Moderate

20% and above: High

Heritability (broad-sense) $h_{(b)}^2$

Heritability in broad-sense was calculated by using the formula proposed by [24]:

$$h^2_{(b)}(\%) = \frac{\sigma^2_g}{\sigma^2_p} \times 100$$

where:

$h^2_{(b)}$ = Heritability (broad-sense);

$\sigma^2_g$ = Genotypic variance

$\sigma^2_p$ = Phenotypic variance.

Heritability percentage was categorized as demonstrated by [25]:

0–30%—Low

30–60%—Moderate

60% and above—High

Expected genetic advance (GA)

Calculated from Allard (1960) at 5 percent selection intensity using the constant 'K' as 2.06.

$$GA = h^2_{(b)} \times K \times \sigma_p$$

where,

GA = Genetic advance;

$h^2_{(b)}$ = Heritability (broad-sense);

$K$ = Selection intensity at 5 per cent = 2.06;

$\sigma_p$ = Phenotypic standard deviation.

Genetic advance expressed as per cent of mean.

The expected genetic advance as expressed in per cent of mean was calculated by the method suggested by [21].

$$GA_{(\%\overline{X})} = \frac{GA}{\overline{X}} \times 100$$

where,

GA = Expected genetic advance;

$\overline{X}$ = Mean of the character under study;

The genetic advance as per cent mean was categorized as suggested by [21]:

0–10%—Low

10–20%—Moderate

20% and above—High

## 3. Results

*3.1. Morphological Variation*

All traits exhibited a wide variation; however, fruit width, %pulp and kernel to nut ratio had the higher spread in their measurements, while shell weight, nut weight and kernel weight had the smaller spread of measurement. Percent pulp (%pulp) measurement had the widest range, between 31 to 97.92, followed by fruit width with a range from 21.8 mm to 69.38 mm and an average of 32.92 cm. Kernel weight had the lowest range, between 1.86 g and 8.16 g, with an average of 5.82 g. All traits studied were significantly different in the genotypes assembled (Table 2).

Heritability estimates for all traits measured were high (Table 3), similarly high genetic advance measurements were also observed (K = 2.06). Most traits recorded a low phenotypic coefficient of variation (PCV), which indicates a possibility of low influence of the environment on the traits. This may be good for selection of traits, as it may indicate stability of traits across several environments. Shell weight had the highest heritability estimate of 0.8, followed by nut weight and %pulp, with heritability estimates of 0.79. Brix had the lowest heritability, of 0.57, of all traits studied. Genetic advance (GA) estimates were very high, especially for pulp weight and shell weight, which were 69.46% and 64.14%, respectively. However, %pulp, which had a very high heritability, had a relatively lower

GA of 25.69% compared to the other traits. Brix, again, had the lowest measurement of 20.91% for genetic advance.

**Table 2.** Trait variation and mean squares.

| Trait | Mean | Range | Mean Square |
|---|---|---|---|
| Fruit weight (g) | 23.05 | 10.25–48.87 | 158.6 *** |
| Fruit length (mm) | 37.89 | 26.44–58.46 | 93.29 *** |
| Fruit width (mm) | 32.92 | 21.80–69.38 | 80.57 *** |
| Brix | 23.11 | 11.93–46.00 | 36.36 *** |
| Pulp weight (g) | 13.94 | 4.67–35.05 | 92.98 *** |
| Nut weight (g) | 5.82 | 2.77–12.10 | 8.35 *** |
| Shell weight (g) | 1.63 | 0.63–4.76 | 1.04 *** |
| Kernel weight (g) | 4.17 | 1.86–8.16 | 4.49 *** |
| %Pulp | 59.38 | 31.00–97.92 | 226.13 *** |
| Kernel to Nut ratio | 71.56 | 51.02–91.75 | 69.85 *** |

*** Significance at **0.001**.

**Table 3.** Genotypic and phenotypic variability, heritability and genetic advance estimates.

| Trait | GV | PV | GCV | PCV | H | GA% |
|---|---|---|---|---|---|---|
| Fruit weight | 48.36 | 61.88 | 30.17 | 34.13 | 0.78 | 54.94 |
| Fruit Length | 27.59 | 35.12 | 13.86 | 16.3 | 0.72 | 24.3 |
| Fruit width | 24.39 | 31.79 | 15 | 17.13 | 0.77 | 27.07 |
| Brix | 9.67 | 17.01 | 13.46 | 17.85 | 0.57 | 20.91 |
| Pulp Weight | 28.32 | 36.33 | 38.19 | 43.25 | 0.78 | 69.46 |
| Nut Weight | 2.56 | 3.23 | 27.51 | 30.9 | 0.79 | 50.45 |
| Shell Weight | 0.32 | 0.39 | 34.77 | 38.82 | 0.80 | 64.14 |
| Kernel Weight | 1.37 | 1.76 | 28.04 | 31.83 | 0.78 | 50.89 |
| %Pulp | 69.29 | 87.55 | 14.02 | 15.76 | 0.79 | 25.69 |

GV = genotypic variance; PV = phenotypic variance; GVC = genetic coefficient of variation, PCV = phenotypic coefficient of variation, H = broad sense heritability; GA = genetic advance.

### 3.2. Multivariate Analysis

3.2.1. Principal Component Analysis

The first component contributed to 46.4% of the observed variation. Of these, fruit weight, width, dry nut weight, shell weight, kernel weight and number of seeds per fruit were the main drivers of the variations (indicated with bold cosines values). Similarly, the second PC contributed 32% of the observed variations, with pulp weight, pulp to fruit weight, kernel to fruit weight, shell to kernel weight and kernel to nut weight being the main factors for the variation (Table 4). All components associated with PC1 were positively correlated, except pulp to fresh weight, which was negative. On the other hand, fruit width, brix, kernel weight, kernel to fresh weight, nut to fresh weight and kernel to nut ratio were all negatively correlated with PC2.

The principal component (PC) biplot shows the relationship between the measured traits and the locations of the collection. The first two components contributed to 78.4% of the total observed variations in the study and were used to generate the biplot.

Shell to kernel ratio, pulp weight, fresh weight and fruit lengths were associated with materials from Bawku, however the association was stronger for pulp weight and fresh fruit weight. Pulp to fresh weight was associated with Navrongo, Wa and Bole materials, while fruit width, number of seeds per fruit, brix, nut to fruit ratio and kernel to fruit weight were associated with materials from Tamale, Yendi and Walewale. Materials from Damongo and Kintampo were not associated with any trait. Materials from Damongo and Kintampo were closely related and so were materials from Yendi and Walewale. Materials from Tamale and Bawku were very isolated from the rest (Figure 5).

**Table 4.** Correlation, quality of representation (Cos2) and contribution of each trait with principal components.

| Trait | Correlation | | Cos$^2$ | | Contribution | |
|---|---|---|---|---|---|---|
| | Dim 1 | Dim 2 | Dim 1 | Dim 2 | Dim 1 | Dim 2 |
| Fresh weight | **0.835** | **0.524** | **0.696** | 0.275 | **10.716** | 6.132 |
| Fruit length | **0.622** | 0.295 | 0.387 | 0.087 | 5.962 | 1.941 |
| Fruit width | **0.854** | −0.247 | **0.730** | 0.061 | **11.234** | 1.360 |
| Brix | 0.391 | **−0.413** | 0.153 | 0.171 | 2.348 | 3.814 |
| Pulp weight | **0.634** | **0.739** | 0.403 | **0.546** | 6.194 | **12.183** |
| Dry nut weight | **0.990** | 0.086 | **0.981** | 0.007 | **15.092** | 0.164 |
| Shell weight | **0.860** | 0.410 | **0.739** | 0.168 | **11.371** | 3.750 |
| Kernel weight | **0.988** | −0.042 | **0.975** | 0.002 | **15.009** | 0.040 |
| Pulp to fresh weight | −0.224 | **0.682** | 0.05 | **0.465** | 0.773 | **10.394** |
| Kernel to fresh weight | 0.520 | **−0.834** | 0.271 | **0.695** | 4.168 | **15.523** |
| Shell to kernel | 0.210 | **0.848** | 0.044 | **0.718** | 0.676 | **16.042** |
| Nut to fresh weight | **0.621** | **−0.715** | **0.386** | 0.512 | 5.941 | **11.432** |
| Kernel to nut ratio | 0.004 | **−0.822** | 0.000 | **0.676** | 0.000 | **15.104** |
| Number of seed/fruit | **0.827** | −0.308 | **0.683** | 0.095 | **10.515** | 2.122 |

Numbers in bold indicate high contribution.

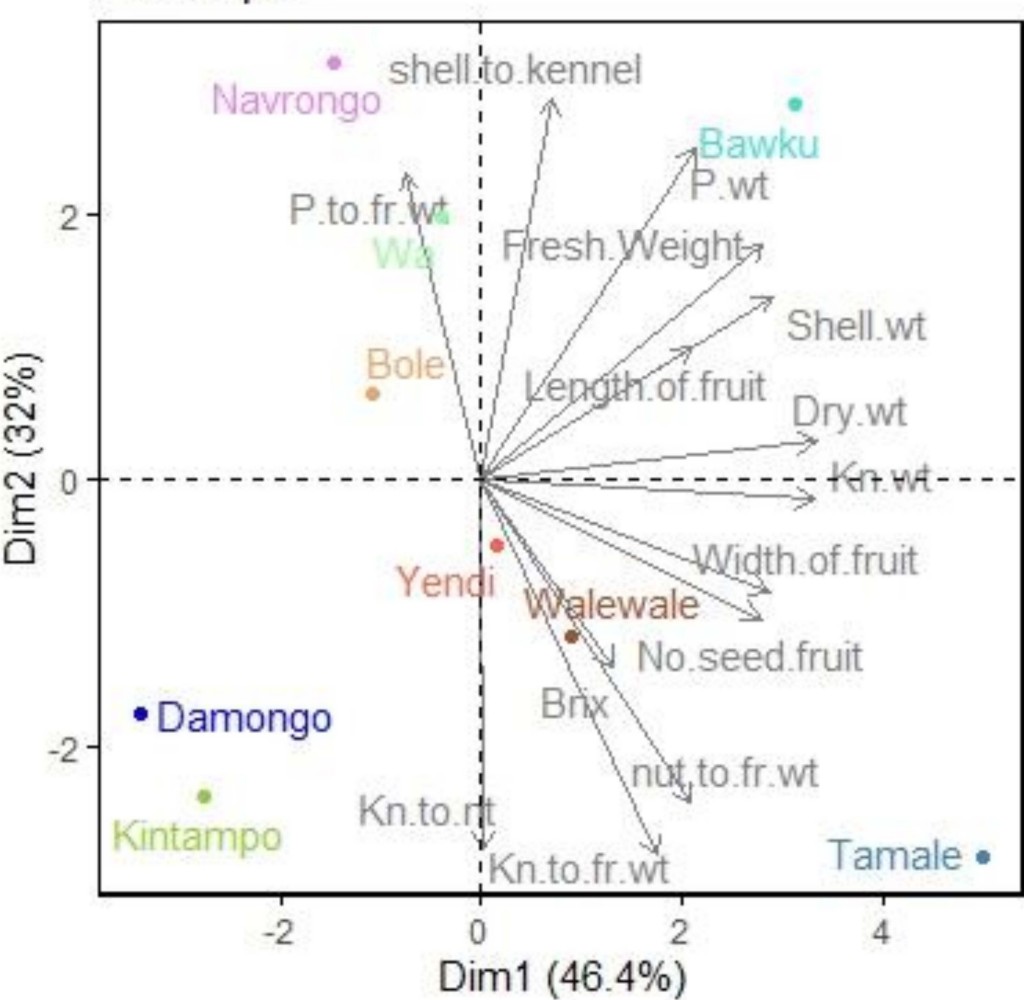

**Figure 5.** Distribution of germplasms with the first two principal components.

### 3.2.2. Morphological Clustering

The relationship between the accessions based on the morphological traits measured was analyzed based on the unweighted pair-group average agglomeration method (Figure 6). At 90%, two main clusters were observed, clusters "A" and "B". Major cluster "A" contained materials from only Tamale. Cluster "B" separated at 80% into two more subclusters, one of its subclusters (B1) contained materials from Bawku, Navrongo and Wa, while the second subcluster (B2) contained five other locations. Navrongo and Wa (cluster D) were related at 25% and a similar relationship was observed between Damongo and Kintampo (cluster E) at 27%. Cluster F had two subclusters with Walewale separating from Bole and Yendi at 45%.

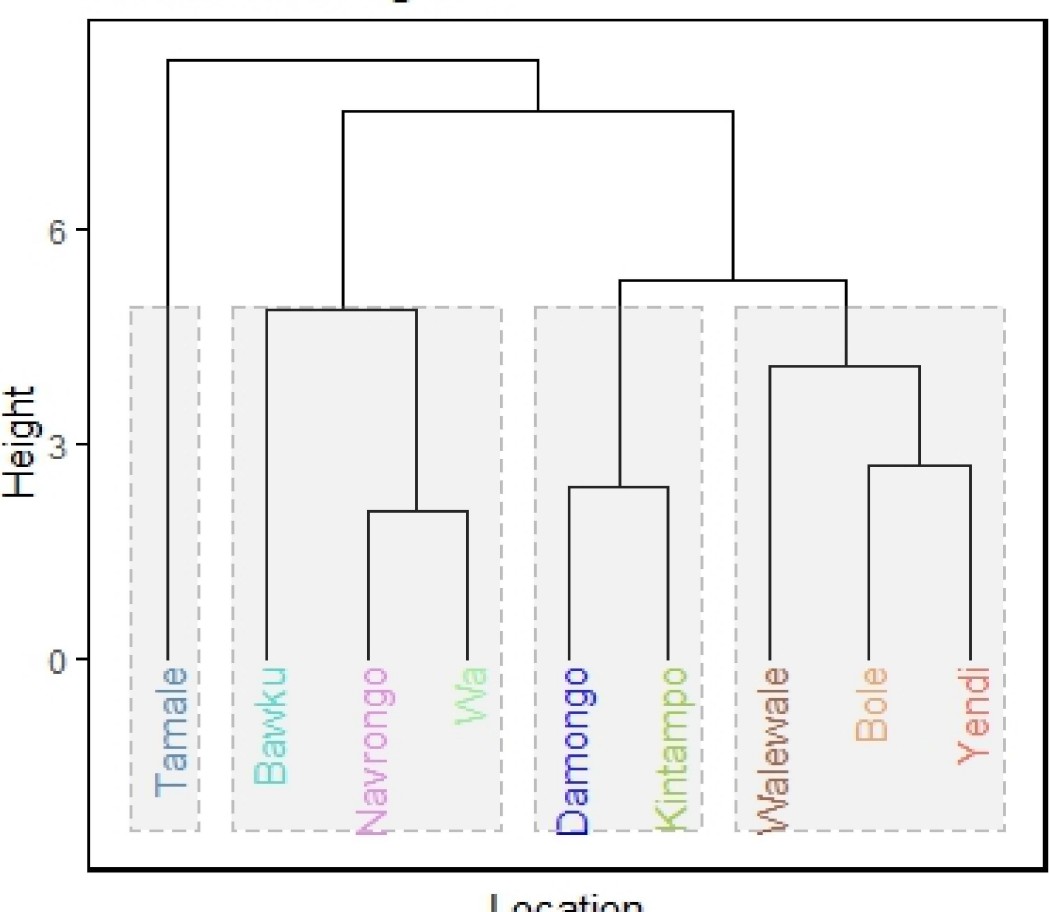

**Figure 6.** Clustering of assembled germplasms based on Pearson's dissimilarity matrix.

### 3.2.3. Correlation among Traits

The pairwise relationships between the traits studied are presented in Table 5. There was a strong positive correlation between fruit weight and fruit length (0.743) and between fruit weight and pulp weight (0.921). Additionally, fruit weight was strongly correlated with dry weight (0.701), shell weight (0.621) and kernel weight (0.678). Brix was not significantly correlated with any of the traits; %kernel weight was significantly correlated to dry to fresh weight ratio. Kernel weight was also significantly correlated to shell weight (0.767). Nut dry weight was also significantly correlated to shell weight (0.875). Pulp weight was highly correlated to fruit weight, length and width.

**Table 5.** Pairwise Pearson correlation coefficients of 12 quantitative variables evaluated on the 95 accessions from 9 locations.

| Variables | FWt | FL | FWd | Brix | PWt | DWt | SWt | KWt | %PWt | %KWt | DW to FW |
|---|---|---|---|---|---|---|---|---|---|---|---|
| NF.L | **0.74** | | | | | | | | | | |
| F.Wd | **0.67** | **0.52** | | | | | | | | | |
| Brix | −0.03 | 0.003 | −0.06 | | | | | | | | |
| Pulp Wt | **0.92** | **0.65** | **0.65** | −0.04 | | | | | | | |
| Dry wt | **0.70** | **0.48** | **0.52** | −0.01 | **0.52** | | | | | | |
| Shell.Wt | **0.62** | **0.40** | **0.39** | −0.12 | **0.47** | **0.88** | | | | | |
| K. wt | **0.68** | **0.48** | **0.51** | 0.02 | **0.50** | **0.97** | **0.77** | | | | |
| %pulp to wt | **0.37** | **0.22** | **0.32** | −0.08 | **0.68** | −0.04 | −0.02 | −0.05 | | | |
| %K to F.wt | **−0.42** | **−0.34** | **−0.23** | 0.08 | **−0.54** | **0.27** | 0.11 | **0.35** | **−0.56** | | |
| Dry to F.wt | **−0.45** | **−0.38** | **−0.26** | 0.05 | **−0.58** | **0.29** | **0.22** | **0.29** | **−0.61** | **0.95** | |
| %K to Nut | 0.01 | 0.04 | 0.07 | 0.11 | −0.01 | 0.06 | **−0.30** | **0.27** | −0.01 | **0.39** | 0.09 |

Values in bold are different from 0 with a significance level alpha = 0.05.

Fruit weight (F.wt); fruit length (FL); fruit width (F.Wd); pulp weight (P wt); nut dry weight to fresh weight ratio (DW to FW); kernel weight (k.wt); percent pulp to fruit weight (Pyield%); percent kernel to total nut weight (%K to Nut)

### 3.3. Qualitative Analysis

Five fruit shapes were identified and classified (Table 6). Ellipsoid shape accounted for 66.5% of the observed fruit shapes, while oblate and oblong accounted for 1% each of the observation. Inverted ovoid shape accounted for 20% of the observation. Fruit surface pubescence was absent in 52.6% of the measurements, while 47.4% had pubescence and were described as rough surface. The fruits were predominantly (96.8%) single-seeded, while ellipsoid seed shape was the most abundant (86.3%).

**Table 6.** Qualitative analysis of fruit characters.

| Variable | Categories | Shape | Counts | Frequencies | % |
|---|---|---|---|---|---|
| Shape of fruit | 1 | Oblate | 1 | 1 | 1.053 |
| | 2 | Spheroid | 8 | 8 | 8.421 |
| | 3 | Ellipsoid | 66 | 66 | 69.474 |
| | 4 | Oblong | 1 | 1 | 1.053 |
| | 6 | Inverted ovoid | 19 | 19 | 20.000 |
| Fruit surface | 1 | Smooth | 50 | 50 | 52.632 |
| | 2 | Rough | 45 | 45 | 47.368 |
| Number of seed(s) per fruit | 1 | | 92 | 92 | 96.842 |
| | 2 | | 3 | 3 | 3.158 |
| Seed shape | 3 | Ellipsoid | 82 | 82 | 86.316 |
| | 6 | Inverted ovoid | 13 | 13 | 13.684 |

% is percentage.

### 4. Discussion

Generally, all quantitative traits measured showed variations among the accessions assembled, most being statistically significant. Similar accounts have been reported in previous studies. There has been a reported variation in fruit length and width [26] and fruit weight, width and length [27]; however, these estimates were higher than those reported in this study. Weight, length and width of fruits harvested from three locations also showed statistical difference [14], just like in this current study; however, their kernel size was much larger than the current study. Fruit weight of shea was reported to be between 10 g to 57 g [28]; however, the range in this study was lower, between 11 g and 44.22 g. Similarly,

ref. [29] observed variations in fruit size and seed number. They reported single seed per fruit (84.66%) as the most common occurrence, which is reaffirmed in this study, with 96.8% of the observation being single seed. Ref. [30] reported on variations in fruit shapes and recorded five fruit shapes, similar to the findings of this study, with the exception being the ellipsoid fruit shape dominating 69.47% of the observations.

Fruit surface pubescence, scored as present or absent on fruits, was 52.6% for fruits without them (smooth fruits) and 47.4% for fruits with these hairs (rough fruits). The purpose of this pubescence has not been studied in shea; however, trichomes on the surfaces of plants in general play a role in protecting them from moisture loss, reflecting high radiation and temperatures and restricting biotic threats such as pathogens, insects and animals [31,32]. A particular type called the non-glandular trichomes are said to vary in morphology, function and size, and are found mainly on plants in dry environments [33]. These recorded variations are very important for shea germplasms conservation, trait discovery, shea improvement and the availability of materials for parkland development through grafting, especially when the crop has seen little to no crop improvement and selection activities over the years.

In order to advance in any crop improvement activity, there is the need to establish that the observed variations are genetically linked and easy to transfer. The heritability estimates for the various traits measured were high, coupled with the genetic advance measurements; this gives a clearer understanding and helps in making accurate and reliable breeding decisions [34]; for instance, kernel weight has a heritability of 0.78, which indicates that 78% of this trait in the collection is genetic-based. In this study, there were both high genetic advance and high heritability estimates for pulp weight, nut weight, kernel weight and fruit weight, which indicates that these traits are under additive gene action and are very ideal for selection. The low phenotypic coefficient of variation (PCV) recorded is a positive for trait, as it may indicate that the environmental effects on the traits are small, suggesting possible stability of the traits across environments. The PCA was able to associate the traits being studied with locations that express the traits the most. This will enable targeted germplasm collection. For instance, per the PCA, Bawku and its environs contains materials with high pulp weight and, by extension, fresh fruit weight. They also had longer fruits and were associated with heavier dry kernel weight too. Hence, for these associated traits, germplasm will be sourced from these locations if we want to breed for those traits. Similarly, Yendi, Walewale and some parts of Tamale were associated with fruit brix; therefore, in breeding programs where we need to improve brix-related traits, our collection will be focused in Tamale.

Relationships between traits are important for selection; this enables sound decisions on which traits could easily be improved without negatively affecting another useful trait. Strong correlations were measured between several traits in the current study; an important one is between kernel weight and fruit weight. In a breeding program to improve kernel size, it will also be easy to select a heavy fruit and still arrive at a good kernel size, since these traits are highly and positively correlated. Pulp weight was highly correlated with most of the fruit parameters, such as fruit weight and fruit length, which indicates that fleshy fruits can easily be selected by just taking the total weight of the fruits or by selecting for fruits with bigger length. The percentage free fatty acids in shea butter have been reported to decrease with increased sugars in shea pulp [27].

Grouping of germplasms allows for proper classification and assigning of appropriate use for each material. It also prevents duplication of materials and injudicious use of resources for their maintenance and conservation. The dendrogram constructed revealed that most of the traits studied were distributed across all the locations of the collection, raising questions on how the species evolved and spread across the entire subregion. It also revealed candidate genotypes for specific crop improvement programs.

## 5. Conclusions

There were variations in traits within the genotypes which are needed for crop improvement. The study also identified locations where particular traits were linked, making trait-targeted collection easy. Most of the traits measured seem to be under additive gene action, due to their high heritability and genetic advance estimates. The fruit and nut shape trait is largely ellipsoid in these materials, which may indicate that this is the predominant shape. However, the availability of other shapes means we can also breed for them. The level of diversity requires that more collections be made within each location; this will result in a richer germplasm collection.

So far, this paper is the first to highlight and estimate the genetic gains and heritability estimates for shea and it offers an insight into the inheritance and gene actions controlling these traits in wild shea accession. This can be used to make breeding decisions and in shea improvement programs.

**Author Contributions:** Conceptualization, W.E.A., M.T.B. and A.D.; Funding acquisition, I.H., F.K.P., M.T.B., A.D. and E.Y.D.; Investigation, W.E.A. and M.T.B.; Methodology, W.E.A., K.O., E.Y.D. and M.T.B.; Writing—original draft, W.E.A.; Writing—review and editing, W.E.A. and S.W.A. All authors have read and agreed to the published version of the manuscript.

**Funding:** Funding for this research was from the National Science Foundation of the United States of America (Award number: 1543942), West Africa Center for Crop Improvement of the University of Ghana and Cocoa Research Institute, Ghana.

**Institutional Review Board Statement:** Not applicable.

**Informed Consent Statement:** Not applicable.

**Data Availability Statement:** Data are available upon reasonable request.

**Acknowledgments:** We are grateful for the hard work and commitment of the field technicians involved in the management and data collection of this trial. We are thankful to Francis Dakura and Bismack Owusu Ansah for their immense contributions. We are thankful to the NSF for their financial support. The assistance of staff of the Cocoa Research Institute, Bole involved in the sample collection is greatly appreciated. This paper is published with the permission of the Executive Director of the Cocoa Research Institute of Ghana.

**Conflicts of Interest:** The authors declare no conflict of interest.

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
