# Peer review of "Heritability and Genetic Advance Estimates of Key Shea Fruit Traits"

_agronomy, doi:10.3390/agronomy13030640_

Round 1

Reviewer 1 Report

Dear authors,

thank you for presented paper. I have some questions and recommendations.

No. of Row/comment

38 and 39 instead of % please use word percent (or describe which shortcut you mean)

51 acording to (2)- unify the formate of citation, better use name of first author

91 wou be nice to have shor summary how many trees were sampled

102 and 103 please check the syntax

123-129 kennel or kernel?

121-129 please check the syntax

Fig.1  better cut of picture 

Fig.2  picture description plese below

Fig.3  better cut of picture 

139-142 maybe change the position of text to Fig.1, please check the syntax

154 genotype? You made genetic fingerprint?

Isnt necessary to present all equations (if you dont change anything in calculations of cited papers).

You calculate narrow or broad sense heritability? (h or H). Have you data from other generations (F1, F1) needet for estimations of heritability. Do you have some genetic data?

Please clear this part of your methods.

Reviewer 2 Report

This is an important and meaningful study.

As the authors mentioned " the genetic erosion  of shea tree has been on-going at an alarming rate" . According to the results of this manuscript, is there some suggestions for this situation? 

It was better to added some more information in the Introduction part, it seemed that this part was too short. Is there some more publications related with your research ?

In Table 1 the location "Wa" is a full name or a abbreviation?

Line 103, "figure 1" should be "Figure 1" what is the number 1, 2, 3, 4, 5 and 99 stand for ? 

Line 110, "figure 3" should be "Figure 3". In Figure 3, some information seemed missing under "Fruit-length".

The format of Table 1 and Line 157 to 158 was different.

Line 278 "RESULTS" or "Results"?

Line 321 "table 4" or "Table 4"? some numbers in this table was marked with bold font, why?

In Table 5, the difference significance analysis should be added.

The conclusion part was too long.

  •  

Reviewer 3 Report

The manuscript “Heritability and genetic advance estimates of key shea fruit traits” evaluated the fruit and kernel diversity of the selected shea, identify the key components contributing to their variation and to assess their genetic basis for crop improvement studies. However, in my opinion, the paper needs minor revisions as suggested below.

1.     Line 13: “Genetic” should be unbold.

2.     Line 51:delete “according to”.

3.     You should use text as the title of the Figure 2.

4.     The first text box has text that is not fully displayed in Figure 3.

5.     Table 2: the “” in the fourth row of the table should be “-”.

6.     Line 314:Delete blank lines.

7.     Table 4:add the explanation of the values in bold.

8.     The title should be unbold in Figure 5.

9.     Line 374-375: this is a form description, please use the correct font size.

10.  Table 6: the format is confusing, it is recommended to rearrange the layout.

11.  References: the serial number of the literature is repeated, please delete the redundant serial number.

Reviewer 4 Report

This paper looks at traits in shea fruits that have been collected from tagged trees that are being used for in situ conservation of the species from a number of different locations. Data has been collected on various fruit and nut characteristics, mostly relating to size and weight. The main aim of the work is to determine the degree of variability in various seed/fruit traits, in order that an evaluation can be made of whether current in situ conservation practices are conserving sufficiency genetic diversity.

In general, it is my opinion that the analysis is overly complex. It has not been made particularly clear how the results of either the heritability analysis or the principal components analysis will be used to make future decisions.

I am not an expert in either heritability or principal component analysis. However, I think it is important that that the authors consider the fact that most of the seeds studied have been collected from a single region (Table 1 shows that of a total of 95 germplasm accessions, over 60 of them have been collected from the same region). Several of the regions from which seeds have been sampled have only one or two accessions. Are there any implications of this for the analysis that has been undertaken? This over-representation of results from a single region should be acknowledged.

In general results have been presented with insufficient explanation of what they mean. The paper is not easy to read and I don’t think that the presentation of results is particularly helpful. For example:

·         Table 2 – the mean squares have been assigned a significance value (*** is significance at 0.001). What does this mean? Line 286 states “there was a significant difference within all traits studied”. What are the traits significantly different from? Was there a significant effect of location? This doesn’t appear to have been calculated, and I believe that such analysis would provide data that is useful. It would be helpful to understand whether certain locations favour different characteristics.

·         There is insufficient explanation of what the values presented in Figures 3 and 4 are. Although a full explanation of how the values in Table 3 have been calculated is given in the methods section, there is no real explanation of what these values mean in terms of the implications for conservation.

·         The first line of the discussion “There were generally variations in all quantitative traits measured with most being statistically different”. Statistically different from what?

·         One of the key factors that is likely to have an affect on seed traits is the environment in which the seeds development. No description of differences in environment between the regions from which seeds have been sampled has been given.

·         It would be useful to know how many trees were typically sampled per population.

Round 2

Reviewer 1 Report

Dear authors team,

thak you for all your corrections in presented paper.

I wish you success in you future scientific work.

Author Response

Thank you

Reviewer 2 Report

I think the current version of this manuscript could be accepted.  

Author Response

Thank you

Reviewer 4 Report

I don’t believe that the authors have improved their manuscript sufficiently to warrant publication. Three out of the four original reviewers recommended that the paper required a major revision before publication, and the changes to the manuscript are only minor. In general there is a requirement for both the purpose of the work done and the main conclusions of the study to be described more clearly.

In response to the particular comments that I raised:

·         It should be made clear how the results of the heritability and PCA would be used. A sentence has been added (L377-379) that states “The PCA was able to associate the traits being studied with locations that express the traits most. This will enable targeted germplasm collection and also target shea materials for parkland development”. It would be beneficial to include an example here – which locations are responsible for the expression of which traits?

·         I didn’t understand the explanation of how specific traits had been determined to be significant (Table 2). Following the author’s response to my comments, I now better understand the work done, but I still don’t think this is clear enough in the manuscript. The mean squares have been assigned a significance value by ANOVA I presume that the authors must have looked at the significance of “genotype” (I presume that “ID” in Table 1 is being considered genotype). L259 states “There was a significant difference within all traits studies”. It needs to be made clear that the traits studied were significantly different in the different genotypes studied.

·         The authors say that over-representation of locations can be overlooked, because the point of the study is to look at how much diversity there is in situ. But surely it is useful to know whether there is greater diversity in certain locations, indeed the comment made about PCA referred to above (L377-379) indicates that this is the case.

·         The explanation that has been given around the values presented Figures 3 and 4 is minimal. In addition to the equations given (or perhaps instead of some of them) I would like to see a more detailed explanation of what the heritabity and genetic advance values that have been calculated mean for future conservation programmes. It would also be helpful to give the values greater context – for a reader who is unfamiliar with these heritability calculations how do we know what high values are? Can other studies be referenced?
